# MorphoGraphX: A platform for quantifying morphogenesis in 4D

**Pierre Barbier de Reuille[1†], Anne-Lise Routier-Kierzkowska[2†], Daniel Kierzkowski[2], George W Bassel[3], Thierry Schüpbach[4], Gerardo Tauriello[5], Namrata Bajpai[2], Sören Strauss[2], Alain Weber[1], Annamaria Kiss[6,7], Agata Burian[1,8], Hugo Hofhuis[2], Aleksandra Sapala[2], Marcin Lipowczan[8], Maria B Heimlicher[9], Sarah Robinson[1], Emmanuelle M Bayer[10], Konrad Basler[9], Petros Koumoutsakos[5], Adrienne HK Roeder[11], Tinri Aegerter-Wilmsen[9], Naomi Nakayama[1,12], Miltos Tsiantis[2], Angela Hay[2], Dorota Kwiatkowska[8], Ioannis Xenarios[4], Cris Kuhlemeier[1], Richard S Smith[1,2]***

[1]Institute of Plant Sciences, University of Bern, Bern, Switzerland; [2]Department of Comparative Development and Genetics, Max Planck Institute for Plant Breeding Research, Cologne, Germany; [3]School of Biosciences, University of Birmingham, Birmingham, United Kingdom; [4]Swiss Institute of Bioinformatics, Lausanne, Switzerland; [5]Chair of Computational Science, ETH Zurich, Zurich, Switzerland; [6]Reproduction et Développement des Plantes, Ecole Normale Supérieure de Lyon, Lyon, France; [7]Laboratoire Joliot Curie, Ecole Normale Supérieure de Lyon, Lyon, France; [8]Department of Biophysics and Morphogenesis of Plants, University of Silesia, Katowice, Poland; [9]Institute of Molecular Life Sciences, Zurich, Switzerland; [10]Laboratory of Membrane Biogenesis, University of Bordeaux, Bordeaux, France; [11]Weill Institute for Cell and Molecular Biology and School of Integrative Plant Science, Section of Plant Biology, Cornell University, Ithaca, United States; [12]Institute of Molecular Plant Sciences, University of Edinburgh, Edinburgh, United Kingdom

**\*For correspondence:** smith@mpipz.mpg.de

[†]These authors contributed equally to this work

**Competing interests:** The authors declare that no competing interests exist.

**Reviewing editor**: Dominique C Bergmann, Stanford University, United States

**Abstract** Morphogenesis emerges from complex multiscale interactions between genetic and mechanical processes. To understand these processes, the evolution of cell shape, proliferation and gene expression must be quantified. This quantification is usually performed either in full 3D, which is computationally expensive and technically challenging, or on 2D planar projections, which introduces geometrical artifacts on highly curved organs. Here we present MorphoGraphX (www.MorphoGraphX.org), a software that bridges this gap by working directly with curved surface images extracted from 3D data. In addition to traditional 3D image analysis, we have developed algorithms to operate on curved surfaces, such as cell segmentation, lineage tracking and fluorescence signal quantification. The software's modular design makes it easy to include existing libraries, or to implement new algorithms. Cell geometries extracted with MorphoGraphX can be exported and used as templates for simulation models, providing a powerful platform to investigate the interactions between shape, genes and growth.

## Introduction

Morphogenesis of multicellular organisms occurs through multiscale interactions of genetic networks, cell-to-cell signaling, growth and cell division. Because of the complexity of temporal and spatial interactions involved, computer simulations are becoming widely used (*Dumais and Steele, 2000*; *Jönsson et al., 2006*; *Sick et al., 2006*; *Lecuit and Lenne, 2007*; *Merks et al., 2007*; *Stoma et al., 2008*;

**eLife digest** Animals, plants and other multicellular organisms develop their distinctive three-dimensional shapes as they grow. This process—called morphogenesis—is influenced by many genes and involves communication between cells to control the ability of individual cells to divide and grow. The precise timing and location of events in particular cells is very important in determining the final shape of the organism.

Common techniques for studying morphogenesis use microscopes to take 2-dimensional (2D) and 3-dimensional (3D) time-lapse videos of living cells. Fluorescent tags allow scientists to observe specific proteins, cell boundaries, and interactions between individual cells. These imaging techniques can produce large sets of data that need to be analyzed using a computer and incorporated into computer simulations that predict how a tissue or organ within an organism grows to form its final shape.

Currently, most computational models of morphogenesis work on 2D templates and focus on how tissues and organs form. However, many patterning events occur on surfaces that are curved or folded, so 2D models may lose important details. Developing 3D models would provide a more accurate picture, but these models are expensive and technically challenging to make.

To address this problem, Barbier de Reuille, Routier-Kierzkowska et al. present an open-source, customizable software platform called MorphoGraphX. This software extracts images from 3D data to recreate curved 2D surfaces. Barbier de Reuille, Routier-Kierkowska et al. have also developed algorithms to help analyze growth and gene activity in these curved images, and the data can be exported and used in computer simulations.

Several scientists have already used this software in their studies, but Barbier de Reuille, Routier-Kierzkowska et al. have now made the software more widely available and have provided a full explanation of how it works. How scientists can extend and customize MorphoGraphX to answer their own unique research questions is also described. It is anticipated that MorphoGraphX will become a popular platform for the open sharing of computational tools to study morphogenesis.

*Kondo and Miura, 2010*; *Varner et al., 2010*; *Kennaway et al., 2011*; *Santuari et al., 2011*; *Aegerter-Wilmsen et al., 2012*; *Kierzkowski et al., 2012*; *Bassel et al., 2014*; *Milde et al., 2014*; *Sampathkumar et al., 2014*; *Yoshida et al., 2014*) in what is now being called *Computational Morphodynamics* (*Chickarmane et al., 2010*). Key to this methodology is the combination of time-lapse microscopy to quantify changes in cell geometry and gene expression with dynamic spatial modeling (*Jönsson et al., 2012*). Confocal microscopy is frequently the tool of choice for data collection, as the proliferation of fluorescence markers and variations in the method make it possible to visualize proteins, organelles, cell boundaries, and even protein–protein interaction and protein movement in vivo. Other technologies such as serial block-face scanning electron microscopy (SEM) (*Denk and Horstmann, 2004*) make it possible to study sub-cellular structures at a much higher resolution on fixed samples. However, despite the rapid advancement of 3D imaging technologies, there is a lack of methods and software to process and quantify these data and to integrate them within simulation environments.

Most simulation models of morphogenesis operate on 2D templates (*Dumais and Steele, 2000*; *Jönsson et al., 2006*; *Sick et al., 2006*; *Merks et al., 2007*; *Stoma et al., 2008*; *Kondo and Miura, 2010*; *Varner et al., 2010*; *Kennaway et al., 2011*; *Santuari et al., 2011*; *Aegerter-Wilmsen et al., 2012*; *Kierzkowski et al., 2012*; *Sampathkumar et al., 2014*). This is not surprising since many key biological processes occur on surfaces, for example in epithelial layers (*Lecuit and Lenne, 2007*; *Savaldi-Goldstein et al., 2007*; *Heller et al., 2014*). Morphogenesis involves complex 3D deformation, such as folding during gastrulation in animal systems or bulging out of new lateral organs in plants, causing significant curvature in the tissues controlling these events. It is therefore essential to be able to quantify cell shapes and fluorescence-based reporters on curved surface layers of cells. The simplest method to achieve this is to take several image slices and project them onto a single plane (*Butler et al., 2009*; *Chickarmane et al., 2010*; *Kuchen et al., 2012*). However, when trying to quantify cell shape change, division orientations, or growth, distortions due to the projection quickly become too large as the angle between the surface and the projection plane increases.

> **Box 1. All resources for MorphoGraphX, including the user manual, the latest software downloads and the source code, can be found on www.MorphoGraphX.org.**
>
> The latest version of the documentation is also distributed with MorphoGraphX itself, and is available from the 'Help' menu (see also *Supplementary file 1*). We encourage users to develop their own plugins to extend MorphoGraphX for new research tasks. If you develop a plugin you think would be of general use, please contact us so that we can include it in the next release of MorphoGraphX. Contact information along with example plugins is provided on the 'Community' tab on the MorphoGraphX website www.MorphoGraphX.org.
>
> DOI: 10.7554/eLife.05864.003

Even small amounts of tissue curvature can hinder the accurate imaging of a single cell layer over an entire sample. To alleviate some of these issues, methods have been developed to determine the 3D position of cell junctions on the surface, while the segmentation into cells is still performed on flat 2D images (*Dumais and Kwiatkowska, 2002*; *de Reuille et al., 2005*; *Routier-Kierzkowska and Kwiatkowska, 2008*). However these approaches are labor intensive, limited to tissues that can be visualized as a flat 2D image, and are not accurate when the angle of the tissue with the projection plane becomes too large. Furthermore, methods based on tissue casts combined with stereo reconstruction of SEM images (*Dumais and Kwiatkowska, 2002*; *Routier-Kierzkowska and Kwiatkowska, 2008*) need to be combined with methods using fluorescent markers (*Uyttewaal et al., 2012*) if gene expression is to be monitored.

Here we present a method and the open-source software MorphoGraphX (www.MorphoGraphX.org, *Box 1*) to quantify the temporal evolution of cellular geometry and fluorescence signal on curved 2D surface layers of cells over multiple time points in both plants and animals. In addition to 2D curved surfaces, MorphoGraphX also possesses a rich set of tools for full 3D image processing and cell segmentation, and can be used to easily transfer realistic cell geometries and fluorescent marker data into computational modeling environments. MorphoGraphX is built from a collection of loadable modules (shared libraries), centered around an interactive visualization core that exploits the latest features of modern Graphics Processing Units (GPUs). This design allows the software to be easily adapted to changing research needs, and facilitates the integration of algorithms from other open-source imaging processing libraries into a custom work flow. The software is the first of its kind specialized to process curved surface layers of cells, and here we demonstrate its capabilities both in plant and animal systems.

## Results

### 3D visualization of voxels and meshes

Modern imaging technologies today provide us with an abundance of data from a variety of sources: Confocal Laser Scanning Microscopy, Magnetic Resonance Imaging and block-face SEM all provide full 3D volumetric data that can be rendered in MorphoGraphX (*Figure 1*, *Video 1*). Our software can also process surfaces, which can be imported from 3D scanners, reconstructions from Stereo-SEM images (*Routier-Kierzkowska and Kwiatkowska, 2008*), focus stacking microscopes and scanning probe methods such as Cellular Force Microscopy (*Routier-Kierzkowska et al., 2012*) (CFM), or extracted within MorphoGraphX from full 3D data sets (*Figure 1*). MorphoGraphX contains a highly optimized rendering engine that is capable of accurate rendering of both semi-transparent surfaces and volumetric data simultaneously. Surfaces are represented by an oriented triangular mesh, which is typically extracted from the surface of an object, and thus represents the outermost tissue layer (*Figure 1A,C,D*), or the boundaries of individual 3D objects (e.g., cells) in the case of full 3D segmentation (*Figure 1B*). Once processed, surfaces and associated data can be exported in a variety of file formats suitable for loading into modeling or analysis softwares, allowing the direct use of sample

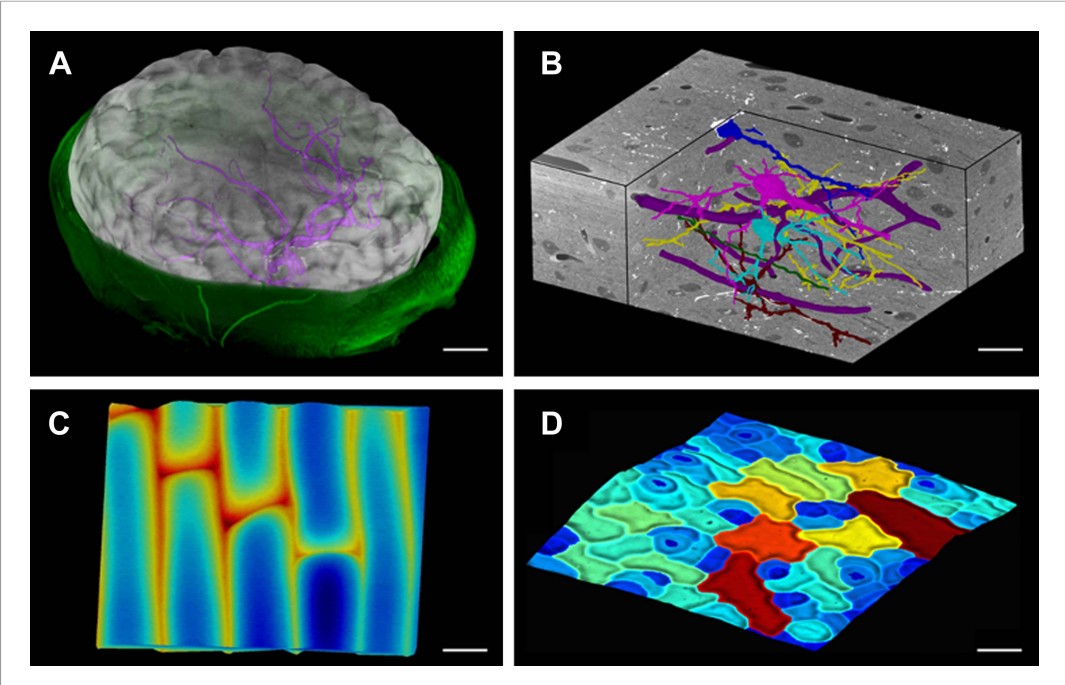

**Figure 1**. MorphoGraphX renderings of 3D image data and surfaces. (**A**) Extraction of a brain surface (gray, semi-transparent surface colored by signal intensity) from a Magnetic Resonance Angiography scan of an adult patient (IXI dataset, http://www.brain-development.org/). Surrounding skull and skin (green) have been digitally removed prior to segmentation. Voxels from the brain blood vessels are colored in purple. (**B**) Serial block-face scanning electron microscopy (SEM) images of mouse neocortex (Whole Brain Catalog, http://ccdb.ucsd.edu/index.shtm, microscopy product ID: 8244). Cutaway view (gray) shows segmented blood vessels (dark purple) and five pyramidal neurons colored according to cell label number. (**C**) Topographic scan of onion epidermal cells using Cellular Force Microscopy (*Routier-Kierzkowska et al., 2012*), colored by height. (**D**) 3D reconstruction of *Arabidopsis thaliana* leaf from stereoscopic SEM images (*Routier-Kierzkowska and Kwiatkowska, 2008*), colored by cell size. Scale bars: (**A**) 2 cm, (**B** and **C**) 20 μm, (**D**) 30 μm.

geometry in computer simulations (*Santuari et al., 2011*; *Kierzkowski et al., 2012*; *Bassel et al., 2014*; *Sampathkumar et al., 2014*; *Yoshida et al., 2014*).

## Feature extraction on curved surfaces

A key strength of MorphoGraphX is the ability to summarize 3D data as a curved surface image. After extracting the shape of an organ, 3D data just below the surface can be projected onto it, creating a curved image of the outer layer of cells (*Figure 2*). This enables the extraction of precise cell outlines without the distortions associated with a flat 2D projection (*Figure 2—figure supplement 1*). We have found that many algorithms designed for 2 and 3D image processing can be adapted to our curved surface images. Feature extraction in MorphoGraphX typically follows a pattern: (i) volumetric data (often a cell outline marker) is pre-processed to remove noise or obstructions; (ii) the object of interest is turned into a mask (binary image); (iii) the object is extracted as a surface mesh; (iv) volumetric data is projected onto the surface; (v) the projection is used for segmentation of the surface into cells (*Figure 2*, *Video 2*). The segmentation can be fully automatic (*Video 3*) or directed by manually placed

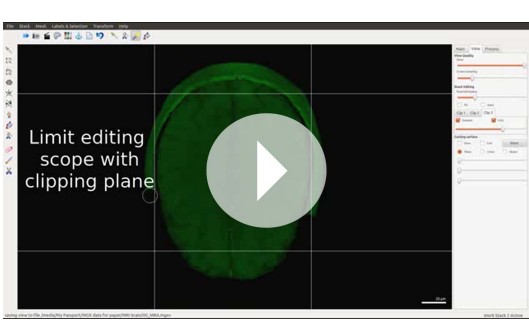

**Video 1.** User interface and rendering in MorphoGraphX.

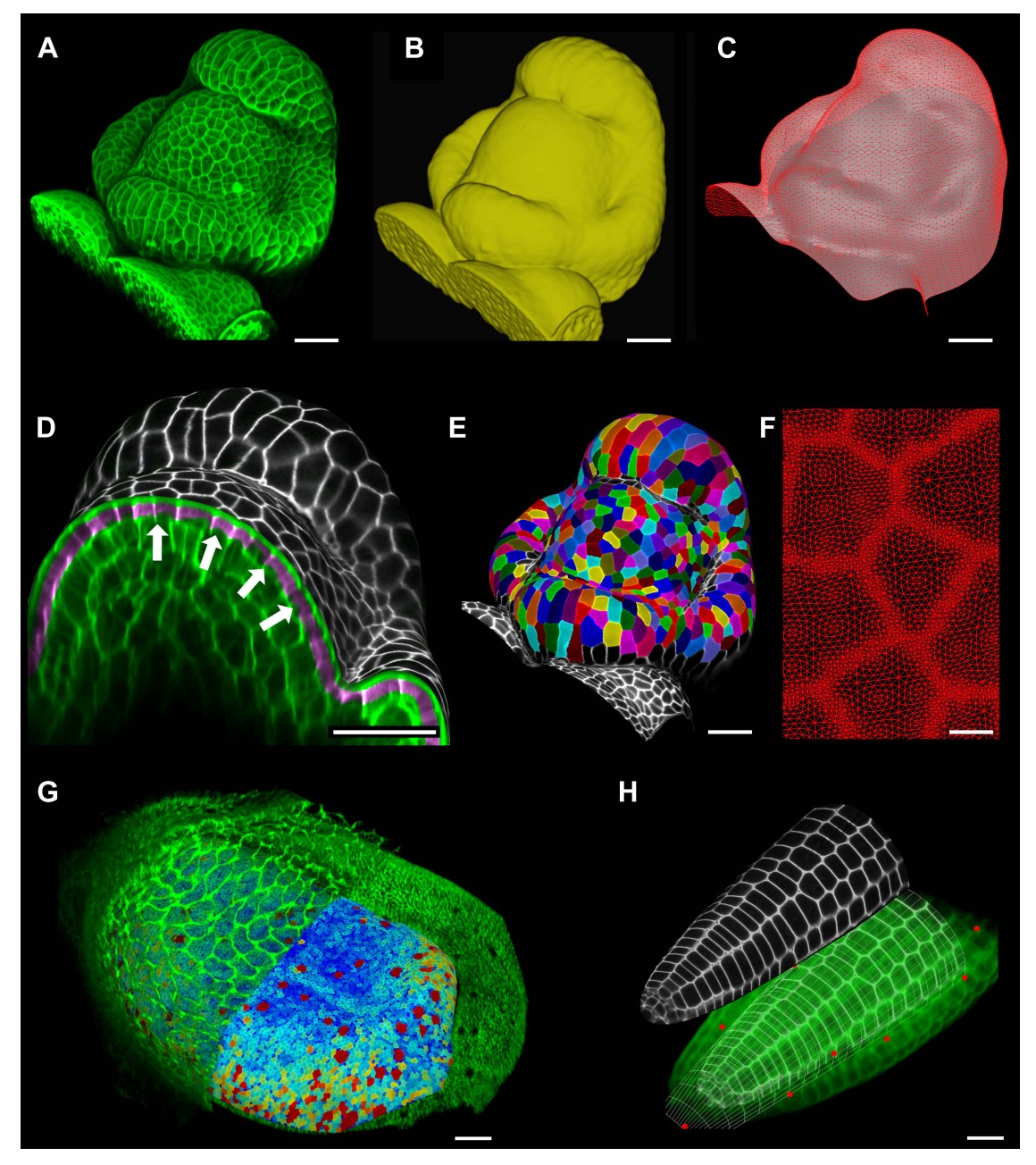

**Figure 2**. Feature extraction and 3D editing of confocal image stacks. A sample workflow from raw data to segmented cells is presented for an *A. thaliana* flower (**A–F**). (**A** and **B**) After removing noise with 3D filters, the stack (green) is converted into a mask using edge detection (yellow). (**C**) A coarse representation of the surface is extracted with marching cubes, then smoothed and subdivided. (**D**) After subdivision, a thin band of signal representing the epidermal layer (purple) is projected onto the mesh, giving a clear outline of the cells. Note that the projection is perpendicular to the curved surface and its depth is user-defined (in this case, from 2 to 5 µm). (**E**) The surface is then segmented with the watershed algorithm, which we adapted to work on unstructured triangular meshes. (**F**) Closeup of adaptive subdivision, with finer resolution near cell boundaries. A similar process flow was used to segment shoot apical meristem in tomato (*Kierzkowski et al., 2012*; *Nakayama et al., 2012*) and *A. thaliana* (*Kierzkowski et al., 2013*), as well as *Cardamine hirsuta* leaves (*Vlad et al., 2014*). (**G**) 3D editing tools can be used to expose internal cell layers prior to surface extraction. Cell shapes extracted from the curved pouch of a *Drosophila melanogaster* wing disc, after removing signal from the overlying peripodial membrane (*Aegerter-Wilmsen et al., 2012*). Alternatively, the stack can be cleaned by removing voxel data above an extracted mesh or conserving only the signal at a defined distance from the mesh, as shown in purple in (**D**) and *Figure 2—figure supplement 2*. (**H**) MorphoGraphX also provides tools to project signal on arbitrary curved surfaces defined interactively by *Figure 2. continued on next page*

*Figure 2. Continued*

moving control points (red). A Bezier surface is highly bent to cut through the cortical cells of a mature *A. thaliana* embryo. Scale bars: 2 μm in (**F**), 20 μm in all other panels.

The following figure supplements are available for figure 2:

**Figure supplement 1**. Maximal projection vs projection of signal on curved surface.

**Figure supplement 2**. Mesh-volume interaction.

seeds. Steps (i–iii) can be repeated as surfaces of interest will often be used to help pre-processing the volumetric data. For example, surfaces can be used to trim the 3D image (*Figure 2—figure supplement 2*), or to select regions of interest for data analysis.

## Interaction with Bezier surfaces

MorphoGraphX allows user-defined surfaces to interact with volumetric data both for visualization and feature extraction. The researcher can interactively define Bezier surfaces to visualize curved slices through an object. By manipulating the Bezier control points it is possible to fit almost any shape to a surface of interest within the sample. An extreme example of this is shown in *Figure 2H* where the surface has been shaped to display the cortical cells of a mature *Arabidopsis* embryo. The Bezier surface can be converted to a triangular mesh, and segmented into cells with the same procedure used for *Figure 2A–E*. The extracted tissue geometry can be then used, for example, as template for simulations (*Santuari et al., 2011*).

## Signal quantification

Once a surface is segmented into cells, data collected simultaneously on a different channel, such as a GFP fusion to a protein of interest, can then be projected onto the segmented surface (*Figure 3*). This allows the quantification of genetic expression and protein localization at the cellular, or sub-cellular scale. As with the cell outlines, the projection creates a curved image of the data that can be processed in a similar way as a planar 2D image. Many tools commonly used for the analysis of flat images (for example in softwares such as Adobe Photoshop, Gimp and ImageJ) have been adapted for use on curved surfaces in MorphoGraphX. This includes Gaussian blur, erosion, dilation, morphological closing, local minima detection, normalization, etc. The flexibility of this approach is demonstrated by our implementation of more complex algorithms, such as the watershed transform for cell segmentation and our adaptation of an algorithm based on signal gradients to compute the orientation of microtubules (*Figure 3A*, *Figure 3—figure supplement 3*) that was previously implemented in 2D (*Boudaoud et al., 2014*).

Signal coming from different tissue layers can be visualized and quantified by adjusting the depth of projection (*Figure 3B–E*). This is particularly useful to distinguish protein expression levels at different depths within an organ. As an illustration, in the shoot apical meristem of *Arabidopsis thaliana* we can observe that the auxin efflux carrier PINFORMED1 (PIN1) is first upregulated in the epidermis at the site of incipient primordium initiation before being activated in deeper layers (*Bayer et al., 2009*; *Kierzkowski et al., 2013*) (*Figure 3C,D* and *Figure 3—figure supplement 1*).

Quantification can also be performed at the sub-cellular scale (*Pound et al., 2012*). The amount of fluorescence signal projected onto the triangle mesh can be divided into a membrane localized portion and a cell interior portion (*Figure 3E* and *Figure 3—figure supplement 2*).

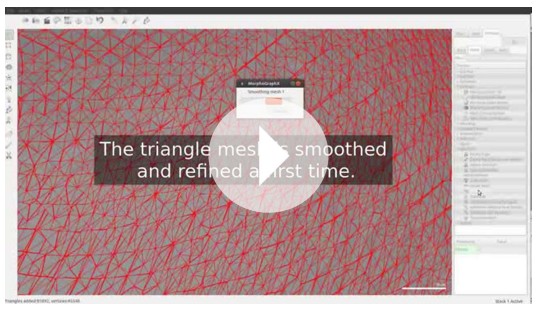

**Video 2.** Manual segmentation of a tomato shoot apex.

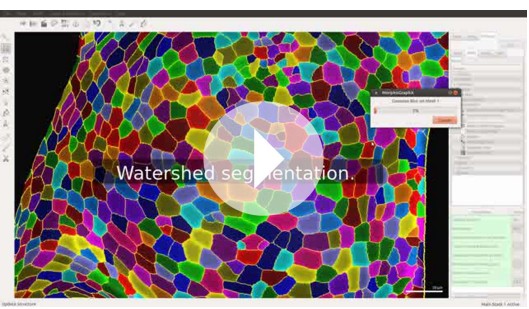

**Video 3.** Automatic segmentation of a tomato shoot apex.

This is accomplished by summing all the signal within a fixed distance from a cell border and considering it as being associated with the membrane, while all the signal further away from the cell outline is called internal. The process can be used to quantify what portion of a tagged protein, for example, the auxin efflux carrier PIN1, is localized to the plasma membrane or internalized (*Nakayama et al., 2012*).

Projection of the signal on the surface allows to summarize essential information from several channels of a large confocal data set into a very compact form. For example, the global shape of the sample can be extracted from an autofluorescence signal, while the cell wall or membrane marker collected in another channel is used to segment cells and obtain their geometry. The expression level of a protein from a third channel may then be quantified at the cellular level based on the segmentation. Finally, several samples in a time lapse experiment can be compared to obtain information about the temporal evolution of shape and gene expression.

## Time lapse analysis

In addition to data from single image stacks, MorphoGraphX is able to process and compare multiple time points. This enables the analysis of stacks before and after experimental treatments, or time-lapse data. This capability relies on an efficient method to co-segment samples from two time points. One approach is to segment both stacks separately and then to use an automated algorithm to match the points (*Fernandez et al., 2010*). However, automatic segmentation and matching can be prone to errors that have to be checked and corrected by hand, which can be very time-consuming depending on the error rate. For this we have developed a user-friendly interface in MorphoGraphX to manually identify cell lineages on curved surfaces representing the tissue at different time points (*Video 4*). Errors in lineage are detected automatically by comparing the neighborhoods of daughter cells and their parents. Once the co-segmentation is complete, changes in cell area or gene expression over the interval between two time points can be computed and visualized as a heatmap (*Figure 4*). Cell proliferation can also be visualized as a colormap (*Vlad et al., 2014*), along with marking of the new walls (*Figure 4*). Pairwise correspondence between time points can be combined in longer time series (*Figure 4—figure supplement 3*), for example to perform clonal analysis over several days (*Vlad et al., 2014*). The data can be output in various formats for further processing, such as the comparison of growth rates with protein expression levels or microtubule orientations.

## Growth directions and anisotropy

In addition to extracting areal growth rates from time-lapse data, MorphoGraphX can also be used to quantify growth directions and anisotropy. The cell junctions (*Figure 4—figure supplement 1*) are used as landmarks to compute the two dimensional Principal Directions of Growth (PDGs) of the surface associated with each cell (*Dumais and Kwiatkowska, 2002*; *Kwiatkowska and Dumais, 2003*; *Routier-Kierzkowska and Kwiatkowska, 2008*). The cell lineage information is used to account for cell division and identify only the junctions that are conserved in between time points (*Figure 4—figure supplement 2*). Principal growth directions and their associated magnitudes can be displayed on the surface of the simplified mesh used for the computation, or stored to be later displayed on the original mesh. The growth anisotropy is computed from the magnitudes of the growth directions (*Figure 4—figure supplement 2*). For visual clarity, growth directions can optionally be displayed only in cells in which the anisotropy is above a user-defined threshold (*Figure 4D*).

Other directional quantities can also be computed, stored and displayed in MorphoGraphX on a cellular basis. For example, the local tissue curvature (*Goldfeather and Interrante, 2004*) can be calculated based on the position of the neighbors closer than a given radius and displayed in a manner similar to the growth directions (*Figure 4B* and *Figure 4—figure supplement 3*), making it a convenient tool for precise staging of fast growing organs such at the shoot apical meristem (*Kwiatkowska and Dumais, 2003*; *Kwiatkowska, 2004*).

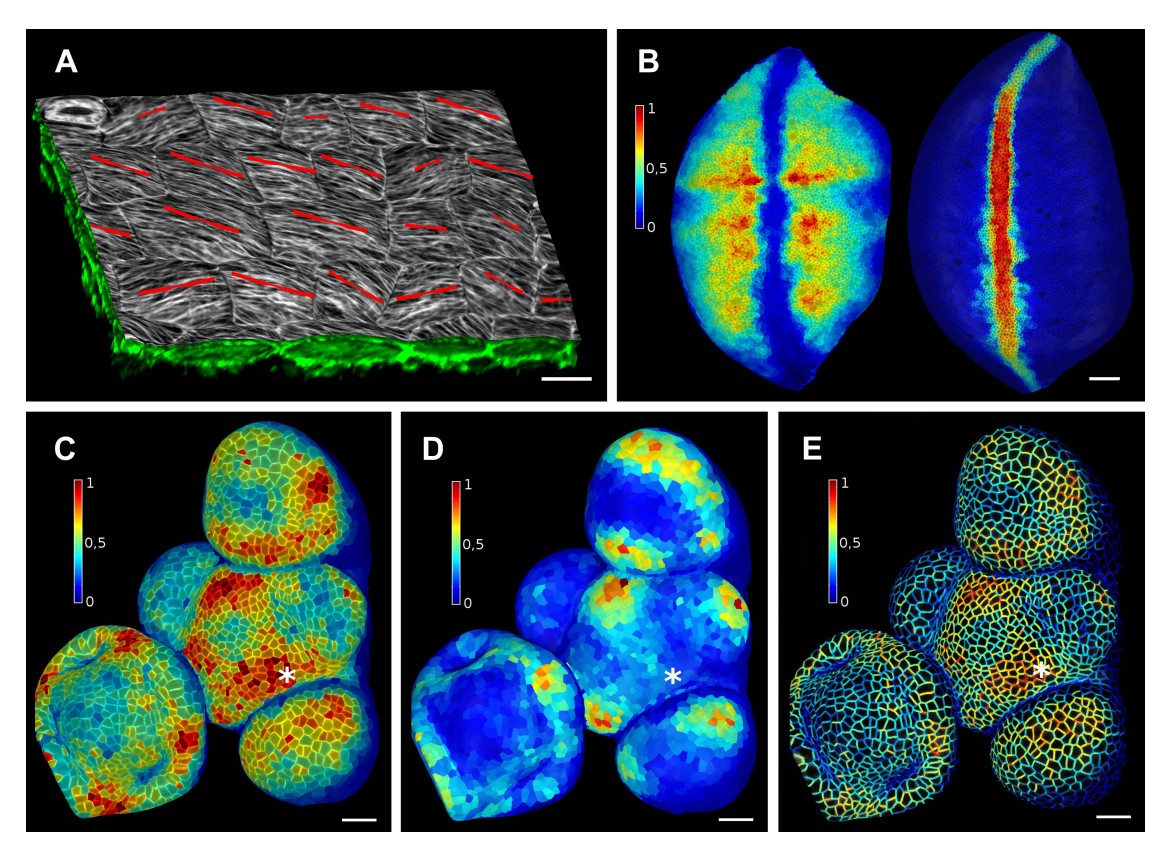

**Figure 3**. Quantification of signal projected on the mesh surface. (**A**) Microtubule orientation (red line) determined in epidermal cells of *C. hirsuta* fruits. Signal for TUA6-GFP (green) at a maximal depth of 1.5 µm was projected on the curved surface and processed with a modified version of a 2D image analysis algorithm (**Boudaoud et al., 2014**) to compute fiber orientation. Line length indicates strength of orientation. (**B**) Quantification of vestigial (left) and wingless (right) transcription in the wing disc of *D. melanogaster* at 0–20 µm depth. Data from (**Aegerter-Wilmsen et al., 2012**). (**C** and **D**) Quantification of PIN1::GFP signal in Arabidopsis shoot apical meristem at different depths. A projection between 0 and 6 µm away from the surface corresponds to the epidermal (L1) layer (**C**), while a depth of 6–12 µm reflects the sub-epidermal (L2) layer. (**E**) Sub-cellular localization of PINFORMED1 (PIN1) in the L1 is assessed by quantification of the projected signal for each cell wall, as in (**Nakayama et al., 2012**). The projected PIN1 signal can be compared with other markers of organ initiation, such as the curvature. While projected PIN1 signal from the L1 (**C** and **E**) shows a clear accumulation of signal at the incipient primordium (star), there is no sign of up-regulation in the deeper layer (**D**) nor visible bulge yet (see *Figure 3—figure supplement 1*). (**C**–**E**) Data from (**Kierzkowski et al., 2013**). Scale bars: 20 µm.

The following figure supplements are available for figure 3:

**Figure supplement 1**. PIN1 expression levels in L1 and L2 vs curvature in Arabidopsis inflorescence meristem.

**Figure supplement 2**. Quantification of PIN1-GFP signal localized to close to the membrane vs internal signal.

**Figure supplement 3**. Quantification of microtubule orientation.

## Growth dynamics of the stem cell niche in the tomato shoot apex

We demonstrate the capabilities of MorphoGraphX by quantifying growth of the stem cell niche and surrounding tissue in the shoot apex of tomato with time lapse imaging over several days (*Kierzkowski et al., 2012*) (*Figure 4* and *Figure 4—figure supplement 3*). The shoot apex is the source of all the aerial structure of the plant. At the summit, a slow growing central zone harbors the stem cell niche, surrounded by a fast growing peripheral zone where organ initiation occurs (*Steeves and Sussex, 1989*; *Dumais and Kwiatkowska, 2002*). The analysis in MorphoGraphX starts with surface extraction followed by manual or automatic segmentation (*Videos 2, 3*), and lineage matching (*Video 4*) of all of the time points in the series. We observed similar patterns of

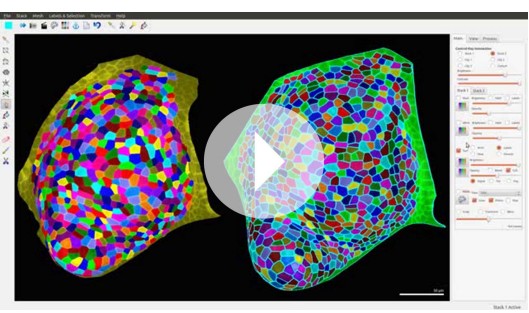

**Video 4.** Lineage tracking and growth analysis of time lapse data on tomato shoot apex.

growth, cell proliferation and organ geometry in the tomato shoot apex as those reported in other species (*Kwiatkowska and Dumais, 2003*; *Grandjean et al., 2004*; *Kwiatkowska, 2004*; *Reddy et al., 2004*; *Kwiatkowska and Routier-Kierzkowska, 2009*). The first geometrical indicator of primordium initiation we noted was a slightly elevated curvature at the corner of the meristem (*Kwiatkowska and Dumais, 2003*; *Kwiatkowska, 2004*). This early change in shape coincided with increased growth in the peripheral zone. The peripheral zone itself displayed differential growth dependent on the dynamics of primordium initiation. Regions adjacent to older primordia exhibited fast, highly anisotropic expansion (*Figure 4* and *Figure 4—figure supplement 3*). In contrast, the part of the meristem closest to the newly separated primordium (P1 in *Figure 4*) was not distinguishable based on growth rates. As previously observed in *Anagallis arvensis* (*Kwiatkowska and Dumais, 2003*; *Kwiatkowska and Routier-Kierzkowska, 2009*), this accelerating growth of the peripheral zone progressively pushed away newly formed organs as they differentiated, making more space available on the meristem for further initiation and suggesting a possible feedback between lateral organ growth and meristem expansion.

In addition to changes in geometry and growth, we used an activity reporter of the growth hormone auxin, pDR5::3xVENUS-N7 (*Heisler et al., 2005*), to follow primordium development. Interestingly, while auxin activity is already visible at the first sign of primordium initiation, DR5 expression does not strictly correlate with growth. In particular, no DR5 signal is detected in the fast expanding regions close to older primordia. We also found that DR5 expression is present in the crease separating young primordia from the meristem, an area where the cells exhibited a slight decrease in surface area (*Figure 4D*). As shown in previous studies (*Kwiatkowska and Dumais, 2003*; *Kwiatkowska, 2004*; *Kwiatkowska and Routier-Kierzkowska, 2009*), the quantification of growth anisotropy shows that cells in the boundary displayed a small increase in length only in the direction parallel to the border between meristem and primordium, suggesting compression by the growing organ (*Hamant et al., 2008*) (*Figure 4D* and *Figure 4—figure supplements 2, 3*).

## 3D cell segmentation and analysis

The extraction of cellular 3D shape is of paramount importance for different purposes, such as to study volumetric deformation, quantify fluorescence expression in 3D, or generate cellular templates for 3D simulation models (*Bassel et al., 2014*; *Yoshida et al., 2014*) (*Figure 5D*). However, volumetric segmentation requires very high quality of signals, since the cell outlines must be visible from all angles. For plant tissues, which often display autofluorescence, 3D segmentation of cells from confocal images is therefore mainly used in the case of cleared, fixed samples (*Bassel et al., 2014*; *Yoshida et al., 2014*) (*Figure 5B–D*) or limited to the outermost layers of cells (*Figures 5A, 6D*). The penetration of confocal images for 3D segmentation of live samples could be improved by using multi-photon confocal microscopy. Another possibility is to combine confocal stacks acquired from different angles (*Fernandez et al., 2010*). Currently it is possible to assemble data from multiple angle acquisition within MorphoGraphX.

MorphoGraphX uses the auto-seeded, morphological watershed algorithm available in the Insight Segmentation and Registration Toolkit (*Yoo et al., 2002*) (ITK) for 3D segmentation. We have developed a collection of user-friendly 3D voxel editing tools allowing the researcher to correct segmentation errors. Alternatively, cells and other objects which are not in contact with each other can be segmented by extracting the surface of the individual objects (*Figure 1B*, *Video1*). As with 2D surfaces of cells, geometrical properties (surface area, wall length, volume) and fluorescent signal (e.g., total signal per cell, membrane localization) of the 3D cells can be quantified (*Figure 5*) and exported to spreadsheet files for further analysis (*Bassel et al., 2014*). Cells segmented in 3D can also be exported for use in simulation models, where highly realistic geometries are required (*Figure 5D*).

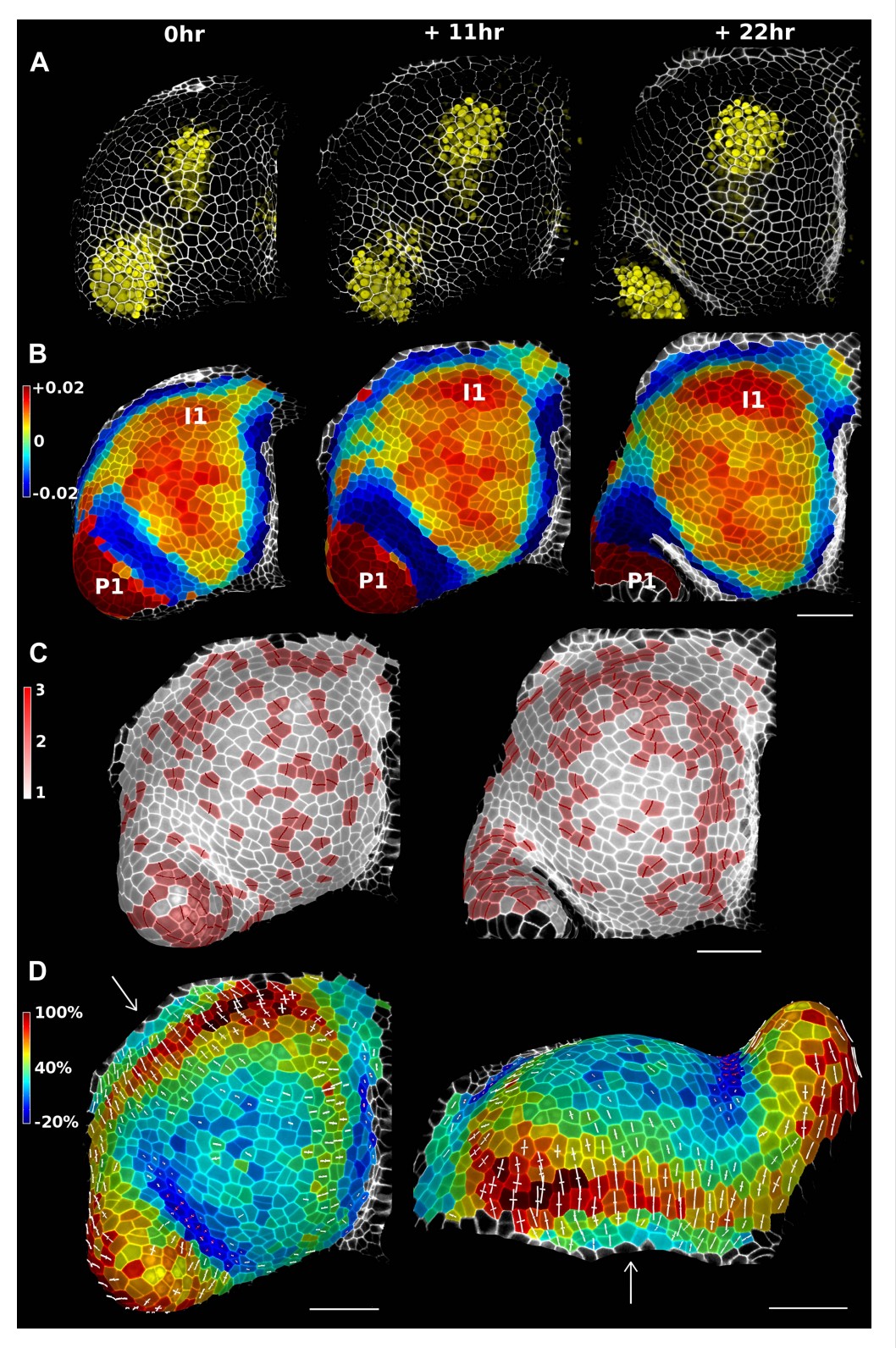

**Figure 4**. Growth in the tomato shoot apex over 22 hr. (**A**) Expression of the auxin activity reporter pDR5::VENUS visualized underneath the semi-transparent mesh. (**B**) Average curvature (µm$^{-1}$) for a neighborhood of 20 µm, with positive values in red, and negative values in blue. (**C**) Shoot apex surface colored by cell proliferation rate as in (*Vlad et al., 2014*). New cell walls are indicated in dark red. (**D**) Top and side views of the heat map of areal

*Figure 4. continued on next page*

*Figure 4. Continued*

expansion over the first 11 hr interval. Principal directions of growth (PDGs) are indicated for cells displaying an anisotropy above 15%, with expansion in white and shrinkage in red. Note the rapid anisotropic expansion of the developing primordium P1 and of the peripheral zone close to the incipient primordium I1, while cells in the boundary between P1 and the meristem contract in one direction (red lines). Arrows indicate the correspondence between top and side views. Raw confocal data from (*Kierzkowski et al., 2012*). Scale bars 50 µm.

The following figure supplements are available for figure 4:

**Figure supplement 1**. Simplification of mesh.

**Figure supplement 2**. Computation of PDGs in case of anisotropic deformation.

**Figure supplement 3**. Analysis of time lapse series of tomato shoot apical growth over 48 hr (5 time points, 12 hr intervals).

## Validation of the method

When projecting data onto surface meshes several sources of error should be considered. Since the Z dimension in confocal images is typically considerably lower in resolution than in XY, it is possible that the view angle affects the results. To estimate the error introduced by this effect, we imaged the same sample twice from different angles (*Kierzkowski et al., 2012*). Co-segmentation with approximately 30° difference in view angle lead to small segmentation differences, averaging to less than 2% (*Figure 6A,B*). Note that there is no obvious bias from the view angle.

Another potential source of error comes from representing 3D cells as a 2D surface. To estimate the error introduced by this abstraction, we co-segmented two time points of growth in the same tomato shoot apex as that shown in *Figure 4*. The cells were segmented on the curved 2D surface, and the process was repeated from the same sample by segmenting the surface layer of cells in full 3D. The heat maps of volume increase in 3D show the same areas of slow and fast growth as the surface segmentation. In cases when the tissue thickness is preserved over growth, as in the epidermal layer of the shoot apex, tracking cell expansion on the surface is therefore a reasonable approximation for volumetric cell expansion (*Figure 6C,D*).

MorphoGraphX offers the possibility to segment cells automatically (*Video 3*) or to place the seeds for watershed segmentation manually (*Video 2*). While automatic segmentation is faster in the case of high quality data, manual seeding is recommended in regions where part of the signal is too faint or blurry, partially masking the cell outline. To estimate the error in our auto-segmentation method, we compared the number of cells segmented automatically vs manually on the same region of two high quality samples. For a tomato shoot apex sample, the auto-segmentation error rate was about 2%, with only 12 cells under-segmented (fused) and 1 cell over-segmented (split) over a total of 576 cells (*Figure 6E*). Once detected, segmentation errors can be easily fixed by the researcher (*Video 3*). Automatic seeding considerably shortens the time needed to segment large samples, such as a *Drosophila* wing disc (*Figure 6F*). The total number of cells varied by about 3% (6304 autosegmented vs 6510 manually seeded cells).

## Discussion

A key strength of our MorphoGraphX software is the ability to accurately extract curved surface meshes from 3D volumetric data and perform image processing on the resulting curved (2.5D) surface images. This has wide application, since many biological processes happen on surfaces, and the method has been proven in both animal (*Aegerter-Wilmsen et al., 2012*) and plant (*Santuari et al., 2011*; *Chitwood et al., 2012*; *Kierzkowski et al., 2012*; *Nakayama et al., 2012*; *De Rybel et al., 2013*; *Kierzkowski et al., 2013*; *Wabnik et al., 2013*; *Sampathkumar et al., 2014*; *Vlad et al., 2014*; *Yoshida et al., 2014*) systems, in embryonic as well as mature tissues. The method is especially powerful for time-lapse confocal imaging, where laser exposure has to be kept to a minimum, limiting penetration to the outermost layers of the sample. In addition to curved surface image processing, MorphoGraphX provides an intuitive and user-friendly interface for the visualization and editing of 3D volumetric data, making it possible to digitally remove obstructing objects from the surface of

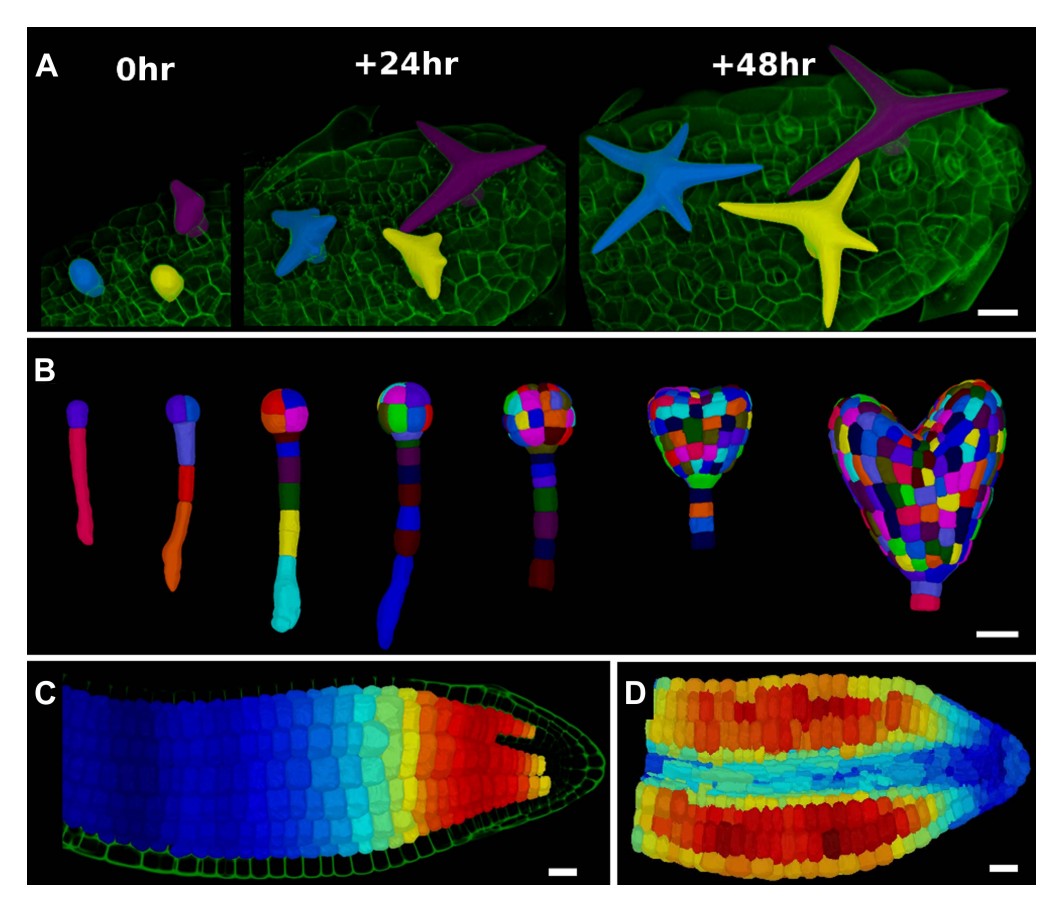

**Figure 5**. 3D segmentation for growth tracking and modeling templates. (**A**) Volume segmentation of trichomes from time-lapse confocal imaging in *Capsella rubella* leaf colored by cell label number. (**B**) Full 3D segmentation of developing Arabidopsis embryos, colored by cell label number. Data from (*Yoshida et al., 2014*). (**C**) False colored projection of the average growth rate along the main axis of an Arabidopsis embryo. Data from (*Bassel et al., 2014*). (**D**) Mechanical model of embryo based on a 3D mesh showing cell wall expansion due to turgor pressure, as published in (*Bassel et al., 2014*). Scale bars: 20 µm.

interest, such as the peripodial membrane overlying the *Drosophila* wing disc (*Aegerter-Wilmsen et al., 2012*). We have also included a range of standard 3D image processing tools, similar to those available in many other softwares (*Fernandez et al., 2010*; *Peng et al., 2010*; *Sommer et al., 2011*; *Federici et al., 2012*; *Mosaliganti et al., 2012*; *Schmidt et al., 2014*). These can be used for 3D segmentation (*De Rybel et al., 2013*; *Bassel et al., 2014*; *Yoshida et al., 2014*), or to pre-process data before surface extraction.

The modular design of MorphoGraphX allows the integration of existing libraries and the creation of custom processing 'pipelines', going from the raw microscopy image to feature extraction and fluorescence quantification. MorphoGraphX is implemented as a collection of shared libraries, and new libraries can be added or removed without recompiling MorphoGraphX. This combines the functionality of plugins with the computational efficiency of C++. The most common operations for 3D visualization, filtering and editing have been written to exploit the massively parallel architecture of modern graphics cards, which can have thousands of processing cores. As a result, 3D operations that would normally be very slow to run on a common PC take seconds to perform, making use of the computational power of inexpensive consumer graphics cards. Many of the more complex operations use the multi-core capabilities of the CPU. This makes most operations interactive and user-friendly, allowing the researcher to easily experiment with new work flows, algorithms and parameters. The flexibility of MorphoGraphX also simplifies the development of modules to import 3D voxel data

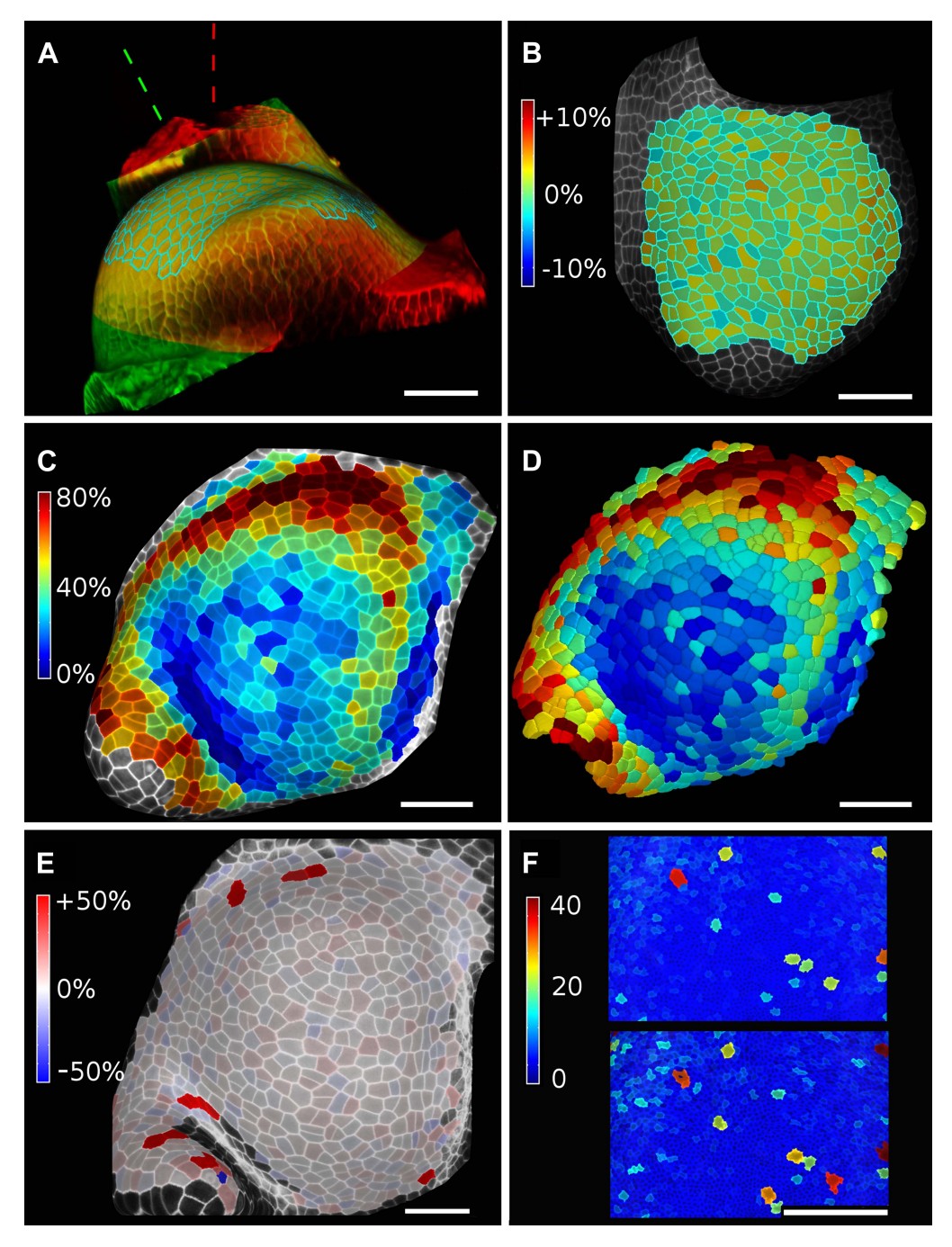

**Figure 6**. Validation of the method. (**A** and **B**) Control for viewing angle. (**A**) A shoot apex imaged from different directions. A first image stack (in red) was acquired before tilting the Z axis (dashed lines) by approximately 30° and acquiring a second stack (in green). Cells were then segmented on both stacks and their areas compared (**B**). Note that the pairwise cell size differences are random, with no obvious trend related to the viewing angle. Average error per cell is less than 2%. Colorbar: relative surface area increase in percent. Panels (**A**) and (**B**) adapted from Figure 5 of *Kierzkowski et al. (2012)*. (**C** and **D**) Comparison between projected areas and actual 3D volumes. (**C**) The epidermal cells of the apex were projected on the surface and segmented. Heatmap shows percent increase in area over 11 hr of growth. (**D**) The same data was segmented in 3D. Heatmap shows the percent increase in volume of cells, same color scale as in (**C**). Note the close correspondence in cell expansion extracted from surface and volumetric segmentations. (**E**) Difference in size between automatically and manually segmented cells on a tomato shoot apex. Cells fused by auto-segmentation are in bright red, split cells are in dark blue. (**F**) Cell sizes (in μm²) from manual (top) and automatic (bottom) segmentation on a fragment of Drosophila wing disc. Scale bars: 40 μm.

and cellular or surfaces meshes from other custom imaging platforms. Such bridges have been created to import data from recently published growth tracking softwares including the MARS-ALT multi angle reconstruction pipeline (*Fernandez et al., 2010*), and the stereo SEM reconstruction software (*Routier-Kierzkowska and Kwiatkowska, 2008*).

MorphoGraphX has been used to quantify cell size (*Aegerter-Wilmsen et al., 2012*), growth and proliferation (*Kierzkowski et al., 2012*; *Vlad et al., 2014*), mechanical properties (*Kierzkowski et al., 2012*) and protein localization (*Nakayama et al., 2012*; *Kierzkowski et al., 2013*), as well as 3D cell geometry (*De Rybel et al., 2013*; *Bassel et al., 2014*; *Yoshida et al., 2014*). In addition to quantification, a current challenge in understanding development is to integrate these new data with computational models. Cellular geometry extracted from biological samples can be easily exchanged with modeling tools, such as Organism (*Sampathkumar et al., 2014*) and VVe (*Bassel et al., 2014*). Meshes extracted in MorphoGraphX can be used directly for realistic simulation templates (*Bassel et al., 2014*), or simplified depending on modeling requirements (*Santuari et al., 2011*; *Wabnik et al., 2013*; *Sampathkumar et al., 2014*). Examples include simulation models of hormone transport (*Santuari et al., 2011*; *De Rybel et al., 2013*; *Wabnik et al., 2013*), cell division plane analysis (*Yoshida et al., 2014*) and 3D cellular models of tissue mechanics (*Bassel et al., 2014*; *Sampathkumar et al., 2014*).

MorphoGraphX was developed by researchers and designed to be easily adaptable to new research requirements. Its user interface was built in close collaboration with experimentalists, with features and techniques added to address research problems and bottlenecks in work flows as they have arisen. Fully automatic tools are complemented with intuitive methods for interactive correction (*Peng et al., 2011*) and validation, greatly increasing the utility of new and existing algorithms.

Streamlined data exchange with modeling tools allows cell geometry and gene expression data to be used as model inputs, and facilitates the validation of simulation results. These features combine to make MorphoGraphX a significant step towards an interdisciplinary computational morphodynamics platform to study the interactions between growth, mechanics and gene expression.

## Materials and methods

### Raw data acquisition

Live confocal time-lapse series of developing flower of *A. thaliana* Col-0 (*Figure 2A–F* and *Figure 2—figure supplement 2*), shoot apical meristem of tomato (*Solanum lycopersicum*) DR5 reporter line (*Shani et al., 2010*) (*Figure 4—figure supplement 3*) and leaf trichomes of *Capsella rubella* (*Figure 5A*) were acquired using SP8 or SP5 Leica confocal microscopes, as described previously (*Kierzkowski et al., 2012*; *Vlad et al., 2014*). After dissection samples were stained with 0.1% propidium iodide (PI) and grown in vitro on medium (*Bayer et al., 2009*). Confocal imaging was performed with a 63× long distance water immersion objective and an argon laser emitting at the wavelength of 488 nm. PI signal was collected at 600–665 nm. In the case of tomato shoot apex, pDR5::3xVENUS-N7 signal was also collected, at 505–545 nm. Distance between stacks was 0.5 µm. Time intervals were 11 hr for tomato and 24 hr for *A. thaliana* and *C. rubella* time lapse series.

Mature *A. thaliana* embryos (*Figure 2H*) were fixed and stained as previously described (*Bassel et al., 2014*) and imaged using a Zeiss LSM710 confocal microscope with a 25× oil immersion lens. Confocal stacks of microtubule marker line TUA6-GFP (*Ueda et al., 1999*) in live *Cardamine hirsuta* fruits (*Figure 3A*) were acquired using a SP2 Leica microscope, with a 40× long working distance water immersion objective and an argon laser emitting at 488 nm. GFP signal was collected at 495–545 nm. The z step between stack slices was 0.2 µm.

The sequential replica method (*Williams and Green, 1988*) was used to acquire a stereopair of SEM images from an *Arabidospsis* leaf surface (*Figure 1D*) as described in (*Elsner et al., 2012*). Stereoscopic reconstruction (*Routier-Kierzkowska and Kwiatkowska, 2008*) was then performed for the stereo pair and converted into a triangular mesh using a custom MorphoGraphX module. All other data presented in this manuscript were acquired for previously published work or available through on-line catalogs.

### Software design and data representation

MorphoGraphX is written in C++ and has been developed on GNU/Linux. For GPU processing, MorphoGraphX uses CUDA (https://developer.nvidia.com/cuda-zone) via the Thrust template library

(http://thrust.github.io). Multi-threaded host processing is done using OpenMP (http://openmp.org/wp/). CUDA requires a compatible nVidia (http://www.nvidia.com) graphics card. The user interface is designed in Qt4 (http://qt-project.org/), and OpenGL is used for 3D rendering (http://www.opengl.org).

MorphoGraphX can be extended using either C++ modules or Python scripts. C++ modules can be loaded at the start of MorphoGraphX through a plug-in system, inspired by the shared library loading architecture of Lpfg in VLab (*Federl and Prusinkiewicz, 1999*). C++ processes can access all the internal data structures used in MorphoGraphX and modify them as needed. They can also call other processes or query for their existence, and get parameter values in a uniform way from the graphical user interface. The last parameter values used for each process are stored in the project (.mgxv) file for future reference. All process calls and their parameters are logged to a re-playable python script log file created in the current directory. Each process is represented as a light C++ object defining the name, parameters and code of the process and is bundled in shared libraries for easy distribution. The shared library is placed into a system or user process folder, and the processes it contains are loaded upon startup.

Python scripts can also be written and executed within MorphoGraphX using the Python Script process. This option offers a more limited interaction with MorphoGraphX as a script is only able to launch other processes and not directly interact with the data structure. However, it allows use of the wealth of modules existing for Python 2.7 for file interactions and data analysis. Most data analysis processes import/export their data as CSV files to facilitate the writing of Python modules for complex or ad-hoc data analysis.

Surfaces are represented by vertex–vertex systems (*Smith et al., 2004*), which implement graph rotation systems. Properties can be stored in the mesh, such as the label attributed to an individual vertex, the normal associated to it or a value for the projected signal. The rendering uses a modified front-to-back depth peeling technique (*Everitt, 2001*) interweaving the volumetric rendering between peels of translucent surfaces. The volumetric rendering itself is done using volume ray casting (*Levoy, 1990*), using the depth of the successive pair of peels to limit the ray casting to the region currently being rendered. This method allows for correct polygon–polygon and polygon-volume intersections. Combined with occlusion detection, we implemented early ray termination when the total opacity of the current fragment becomes too high for subsequent peels to be visible.

MorphoGraphX can be easily extended to import and export voxel and triangle mesh data in various formats. For voxel data, MorphoGraphX can read and write the tiff format compatible with ImageJ or Fiji (*Schindelin et al., 2012*). 3D data can also be loaded from series of 2D images using any of the various image formats supported by the C++ Template Image Processing Toolkit (CImg) (*Tschumperlé, 2012*). The Visualization Toolkit (VTK) (*Wills, 2012*) is used to import and export VTK triangle meshes. Various other formats, such as the Stanford Polygon File format (.ply), Wavefront's Object format (.obj) or 3D Systems StereoLithography format (.stl), are also supported directly. For many of the mesh imports, polygons with be converted to triangles upon loading by generating a center point and making a triangle fan.

## Feature extraction from volumetric data

The first step in processing the data stacks is to remove noise and then identify the which voxels belong inside of the organ (*Figure 2A,B*). 3D image processing filters for noise reduction are followed by edge detection combined with feature filling. Once the inside of the organ is identified it is represented as a binary image (*Figure 2B*). Next the surface is extracted using a variant of the marching cubes algorithm (*Bloomenthal, 1988*). Fairly large cubes are used, creating a relatively coarse mesh and avoiding the extraction of very small features due to surface noise (*Figure 2C*). Once a coarse surface mesh is extracted, it is uniformly subdivided. The resolution of this initial mesh has to be sufficient for a first segmentation, which can be subsequently refined.

After the surface is extracted and subdivided, a column of signal normal to the surface is projected onto the mesh at every vertex, creating a 2D curved image of the cell outlines on the surface layer (see *Figure 2D*, *Video 2*). The image is segmented into cells using a seeded watershed segmentation algorithm. After blurring the image, auto-seeding is performed by finding local minima of signal within a given radius. Seeds are then propagated with watershed. Depending on the radius used for detecting the local minima, several seeds can be placed within a single cell, resulting in over-segmentation. The cells are later merged, based on the relative strength of signal on the walls separating them (*Video 3*). Normalization of the signal with a radius greater than that of the largest cell typically improves

merging results. For convenience, the processes are chained together in a single auto-segmentation process. The final segmentation is then manually corrected. The amount of manual correction required can vary depending on signal quality, and in some cases it can be more efficient to perform some or all of the seeding manually.

We have placed emphasis on designing the user interface for MorphoGraphX to streamline the process of manual seeding and segmentation correction (*Videos 2, 3*).

After the initial segmentation, the edges of the cells will often look rough, as there are not enough points to describe them correctly. To extract the geometry more precisely, the mesh can be subdivided specifically at the interfaces between cells (*Figure 2F*) or in areas of high signal intensity. After subdivision the signal is re-projected, and the surface segmented again. The seeds are retained during this process so that re-seeding is not required. Several steps of subdivision and re-segmentation can be applied until the desired precision is achieved (*Video 2*). The resulting mesh will be dense around the areas of interest (e.g., the interface between cells), while keeping the areas of low interest (the inside of cells) coarse, thus limiting the total size of the mesh file.

### Cell growth and geometry analysis

Once the cells have been segmented from two different time points, the cells and their progeny can be identified manually. Each mesh is loaded in a separate channel and roughly aligned manually so that the cells outlines match. For each cell in the second time point, the user identify a mother cell with a mouse click (*Video 4*). The lineage information is then used to compare cell size (areal growth) or the projected signal intensity in the original cells and their daughters.

A segmented mesh contains information about the cells neighborhood, that is, which are the cell walls shared by two cells and where do the cell walls intersect. The mesh can be simplified to contain only vertices necessary to describe each cell contour and the connections between neighbor cells (*Figure 4—figure supplement 1*). Plant cells do not slide with respect to each other, therefore the junction between cell walls can be used as landmarks to track tissue deformation over time series (*Green et al., 1991*). Combined with the cell lineage information, the simplified cellular mesh (*Figure 4—figure supplement 1*) is used to find the correspondence between cell junctions in meshes extracted from different time points (*Figure 4—figure supplement 2*). After identifying pairs of junctions conserved in both meshes using the lineage information, we project for each cell the junctions on the average cell plane and compute a best fit of the 2D transformation (translation, rotation, anisotropic scaling) that will minimize the squared distance between pairs of junctions (*Goodall and Green, 1986*; *Routier-Kierzkowska and Kwiatkowska, 2008*). Decomposing the transformation into singular vectors and values gives the PDGs and associated scaling values (PDG$_{max}$, PDG$_{min}$), that describe anisotropic growth. Anisotropy values used in (*Figure 4* and *Figure 4—figure supplements 2, 3*) were computed according to the following definition: anisotropy = PDG$_{max}$/PDG$_{min}$.

The cellular mesh can also be used to compute other quantities, such as the tissue curvature (*Figure 3—figure supplement 1* and *Figure 4—figure supplement 3*). In that case the vertices belonging to the cell outline are used to compute the principal curvatures for each cell center, within a given periphery. Color maps resulting from the computation of growth, curvature, signal quantification, etc. can be written to a spreadsheet giving easy access for further processing. Similarly, cell axis vectors can also be exported to be either re-rendered in MorphoGraphX or loaded for further analysis using other software, such as Matlab or Python.

### Volumetric (3D) segmentation

The ITK (*Yoo et al., 2002*) auto-seeded watershed segmentation algorithm implemented in MorphoGraphX was used to segment the cells in 3D in *Figures 5, 6D*. After segmentation the cell surface is extracted using marching cubes and labeled. In some cases individual cells can also be segmented using a custom edge detect function from multiple angles (*Figure 1B*, *Video 1*). MorphoGraphX also provides the possibility to stitch stacks or combine multi angle stacks in 3D. However, this is not a pre-requisite for 3D segmentation in MorphoGraphX.

### Acknowledgements

This work was funded by the SystemsX.ch RTDs Plant Growth 1 & 2 (RSS and CK), the SystemsX.ch RTD WingX (PK and KB), the SystemsX.ch RTD SyBIT (IX), the Swiss National Science Foundation

interdisciplinary project grants CR32I3_132586 and CR32I3_143833 (RSS), the Swiss National Science Foundation international short research visit (AB and RSS), Human Frontier Science Program grant RGP0008/2013 (AHKR and RSS), European Research Council Advanced grant (PK), the Biotechnology and Biological Sciences Research Council grant BB/L010232/1 (GWB), University of Birmingham Research Fellowship (GWB), Forschungskredit of UZH (TA-W), the Polish National Science Centre MAESTRO research grant No 2011/02/A/NZ3/00079 (DK), Deutsche Forschungsgemeinschaft grant SFB 680 (MT), Human Frontier Science Program grant RGY0087/2011 (AH), and Max Planck Society W2 Minerva program grant (AH).

## Additional information

### Funding

| Funder | Grant reference | Author |
| --- | --- | --- |
| Schweizerische Nationalfonds zur Förderung der Wissenschaftlichen Forschung | international short research visit | Agata Burian, Richard S Smith |
| Human Frontier Science Program (HFSP) | RGP0008/2013 | Adrienne HK Roeder, Richard S Smith |
| European Research Council (ERC) | Advanced grant | Petros Koumoutsakos |
| Biotechnology and Biological Sciences Research Council (BBSRC) | BB/L010232/1 | George W Bassel |
| University Of Birmingham | Research Fellowship | George W Bassel |
| Universitat Zurich | Forschungskredit | Tinri Aegerter-Wilmsen |
| Narodowe Centrum Nauki | MAESTRO research grant No 2011/02/A/NZ3/00079 | Dorota Kwiatkowska |
| Deutsche Forschungsgemeinschaft | SFB 680 | Miltos Tsiantis |
| Human Frontier Science Program (HFSP) | RGY0087/2011 | Angela Hay |
| Max-Planck-Gesellschaft | W2 Minerva program grant | Angela Hay |
| Schweizerische Nationalfonds zur Förderung der Wissenschaftlichen Forschung | CR32I3_132586 | Richard S Smith |
| Schweizerische Nationalfonds zur Förderung der Wissenschaftlichen Forschung | CR32I3_143833 | Richard S Smith |
| The Swiss Initiative in Systems Biology | RTDs Plant Growth 1 & 2 | Cris Kuhlemeier, Richard S Smith |
| The Swiss Initiative in Systems Biology | RTD WingX | Konrad Basler, Petros Koumoutsakos |
| The Swiss Initiative in Systems Biology | RTD SyBIT | Ioannis Xenarios |

The funders had no role in study design, data collection and interpretation, or the decision to submit the work for publication.

### Author contributions

PBR, PK, CK, RSS, Conception and design, Analysis and interpretation of data, Drafting or revising the article; A-LR-K, Conception and design, Acquisition of data, Analysis and interpretation of data, Drafting or revising the article; DK, AB, HH, MBH, KB, TA-W, Analysis and interpretation of data,

Contributed unpublished essential data or reagents; GWB, Drafting or revising the article, Contributed unpublished essential data or reagents; TS, GT, NB, SS, AW, AK, Contributed program code, Conception and design; AS, Acquisition of data, Analysis and interpretation of data; ML, Contributed program code, Analysis and interpretation of data; SR, Analysis and interpretation of data; EMB, NN, IX, Conception and design, Analysis and interpretation of data; AHKR, Conception and design, Analysis and interpretation of data, Contributed unpublished essential data or reagents; MT, AH, DK, Analysis and interpretation of data, Drafting or revising the article, Contributed unpublished essential data or reagents

**Author ORCIDs**
Adrienne HK Roeder, http://orcid.org/0000-0001-6685-2984

## Additional files

### Supplementary file

• Supplementary file 1. MorphoGraphX User Manual. The MorphoGraphX user manual is written in a tutorial style, and the accompanying data sets are available for download on the MorphoGraphX website (www.MorphoGraphX.org). Installation instructions for MorphoGraphX and troubleshooting tips are in Section 16 towards the end of the manual.

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
