## [Decision Letter]

Thank you for sending your work entitled “MorphoGraphX: A platform for quantifying morphogenesis in 4D” for consideration at *eLife*. Your article has been favorably evaluated by Detlef Weigel (Senior editor) and two reviewers, one of whom is a member of our Board of Reviewing Editors.

The Reviewing editor and the other reviewer discussed their comments before we reached this decision, and the Reviewing editor has assembled the following comments to help you prepare a revised submission.

Overall, there was considerable enthusiasm for the MorphographX resource, but it was felt that its broad utility as a resource is dependent on making several additions, modifications and consolidations to the existing manuscript and accompanying software and documentation.

These additions include detailed technical specifications about how to use MorphoGraphX, its flexibility with other software packages, importing and exporting data analyses, and community-based resources to accommodate the heavy usage, additional resources/packages/libraries, and software updates that will undoubtedly ensue for what will very likely be a popular and useful resource.

Specific issues to address are enumerated below (1-3):

1) Production of a “user manual” with sufficient detail that a new user could install and use the software for the applications described in this manuscript. MorphographX should be described in sufficient detail to be used by developmental biologists without extensive computational backgrounds. The manual could be modeled upon the protocol currently published (and behind a pay-wall) in the recent paper in Plant Cell Morphogenesis Methods and Protocols (2014). All reviewers consider it essential that the “operating instructions” for something touted as open-source software also be made freely available.

1A) The integration into a complete experimental pipeline should be laid out. For example, the manuscript mentions the possibility to export data to other software for modeling or statistical analysis. However the workflow for such a transition is not described. How can data derived from MorphoGraphX be exported and processed with existing software (e.g. R, Excel, modeling software)? Buried in the Materials and Methods is the mention that “Most data analysis processes import/export their data as CSV files…”, but how to do this is never explained in the manuscript. What variables or statistics should the user expect to be exported, how much control does the user have in the types of data exported or how processed it is? Additionally, can this be done through the GUI, or will it be done through the terminal? Maybe an interesting way to demonstrate this is for some of the example data analyzed in this manuscript, provide .csv/.txt example outputs as supplemental information.

1B) On the other end of the pipeline, what kind of data structure is needed as input (are there “ideal” settings for z-steps, fluorescence intensities, time intervals).

2) Access to software, updates for software and the manual must be accessible in the same place. These could be hosted on the *eLife* site or on a third party site, but stable user access must be possible. Readers should know where to go to download the software, manual, and other resources. www.MorphoGraphX.org needs to be mentioned more than once, more prominently in the *eLife* article; creating a box in the *eLife* manuscript with this information prominently highlighted would be one approach.

Currently, if one visits www.MorphoGraphX.org, there are references to publications, to a “protocol” paper, and notes from courses on MorphoGraphX, but no consolidated resource. Having a single resource/manual (rather than several scattered ones) is very important. Having a single resource directs users to one set of instructions, and offers a single starting place for beginners. Additionally, if manual versions are tied to software updates, then documentation is traceable, and over time dedicated users will be able to pinpoint changes quickly that have accumulated since the last version. It may be that the internal documentation for the software is extensive, but a separate document is important still.

The online resource with the actual software to download and manual will be critical. The current hosting at Max Planck is fine (likely to be stable), but have the authors considered someplace like GitHub or other widely used version control websites to keep their software and updates? This may make access more robust in the long term.

Related to documentation: do the authors anticipate that the community will write their own plug-ins, modules, libraries, and code for MorphoGraphX that will be shared? Will it become like ImageJ where the community contributes as much as the software platform? If so, maybe this should be built into MorphoGraphX.org as well.

One way to consider providing documentation follows an example from Hadley Wickham's ggplot2. In addition to the “manual” which is cut and dry about the functions to use, there is a book and also an excellent self-published online resource. Many other R folks also publish on top of this all in the Journal of Statistical Software, or a “vignette”, to give examples of how to use packages (the *eLife* MorphoGraphX article is probably closest to a vignette).

R example of documentation, ggplot2:

The actual manual: http://cran.r-project.org/web/packages/ggplot2/ggplot2.pdf

The online resource: http://docs.ggplot2.org/current/

A book: https://books.google.com/books?id=bes-AAAAQBAJ&printsec=frontcover&dq=ggplot2:+Elegant+Graphics+for+Data+Analysis&hl=en&sa=X&ei=9U6oVJnJKsy8yQTJtoGwAw&ved=0CDIQ6AEwAA#v=onepage&q=ggplot2%3A%20Elegant%20Graphics%20for%20Data%20Analysis&f=false

3) There should be extensions/additions to the capabilities beyond what was published previously by the authors and others. Several of these applications have been hinted at in the text (for example semi-automated segmentation, extraction of 3D data from complex cell shapes (like lobed cells), tracking of subcellular elements). The detailed explanation of how to do one or two of these other analyses should be provided in the revised version.

Our suggestions for those one or two detailed applications are:

3A) Quantification of differential fluorescence levels within a cell (starting third paragraph of the subsection headed “Signal quantification” about PIN1-XFP fluorescence). Such a tool would likely be of general use. Some things to consider: Is it possible to obtain a percentage of membrane area that is occupied by PIN1-XFP compared to the whole cell? How does the software distinguish which of two neighboring cells expresses the signal? Can the software determine the overall orientation of the PIN1-XFP signal, as compared to the general tissue orientation (e.g. can it determine whether PIN1-XFP in root elongation zones always faces tipwards)?

3B) The opportunity for subcellular analysis was hinted at, but more details are needed. For example, quantification of microtubule orientation is an important feature (Figure 3), but it is not explained clearly here. How is this algorithm better than the published or commercially available ones (e.g. [22])? Is it possible to get the standard deviation of the orientation (e.g. the anisotropy of MT orientation) mapped onto a tissue? How might this tool be used to determine microtubule growth/shrinkage rates? Is it adaptable to track particles like vesicles or Golgi stacks? Has the software been used with data that features things moving at high velocities (e.g. captured with spinning disc confocal microscopy)? Where are the limitations of the software in this regard?

---

## [Author Response]

*1) Production of a “user manual” with sufficient detail that a new user could install and use the software for the applications described in this manuscript. MorphographX should be described in sufficient detail to be used by developmental biologists without extensive computational backgrounds. The manual could be modeled upon the protocol currently published (and behind a pay-wall) in the recent paper in Plant Cell Morphogenesis Methods and Protocols (2014). All reviewers consider it essential that the “operating instructions” for something touted as open-source software also be made freely available*.

*1A) The integration into a complete experimental pipeline should be laid out. For example, the manuscript mentions the possibility to export data to other software for modeling or statistical analysis. However the workflow for such a transition is not described. How can data derived from MorphoGraphX be exported and processed with existing software (e.g. R, Excel, modeling software)? Buried in the Materials and Methods is the mention that “Most data analysis processes import/export their data as CSV files…”, but how to do this is never explained in the manuscript. What variables or statistics should the user expect to be exported, how much control does the user have in the types of data exported or how processed it is? Additionally, can this be done through the GUI, or will it be done through the terminal? Maybe an interesting way to demonstrate this is for some of the example data analyzed in this manuscript, provide .csv/.txt example outputs as supplemental information*.

User guide: We developed a user manual that explains the use of MorphoGraphX in sufficient detail that a novice in image processing should be able to reproduce the results in the manuscript. It is written in a tutorial style, and we have provided some sample datasets to go with it. At the end is a reference section that also includes installation instructions, information to get started with plug­in development, and instructions on how to compile from source code.

CSV files: The user guide also includes a detailed description of the format of .csv files that are produced as results of data analyses.

Data exchange file formats: The user guide lists the file formats native to other softwares (i.e. STL, OBJ, ply, tiff, etc.) that are available to export/import data into MorphoGraphX. The guide also explains where to find documentation for the custom file formats used by MorphoGraphX (.mgxm, mgxs) in the on­line programmer (doxygen) documentation.

On-line programmer documentation: Reference documentation for programming plug­ins has been created using Doxygen, and is distributed with MorphoGraphX. This information is accessible from the Help menu from within MorphoGraphX.

The user manual has been uploaded with the submission, however it will also be available on the website and distributed with MorphoGraphX and (under the help menu).

*1B) On the other end of the pipeline, what kind of data structure is needed as input (are there “ideal” settings for z-steps, fluorescence intensities, time intervals)*.

A section describing optimal confocal settings and collection methods is contained in the new user manual.

*2) Access to software, updates for software and the manual must be accessible in the same place. These could be hosted on the* eLife *site or on a third party site, but stable user access must be possible. Readers should know where to go to download the software, manual, and other resources.*
*www.MorphoGraphX.org*
*needs to be mentioned more than once, more prominently in the* eLife *article; creating a box in the* eLife *manuscript with this information prominently highlighted would be one approach*.

We have added a link towww.MorphoGraphX.org in the Abstract and a box to highlight this information in the article.

*Currently, if one visits*
*www.MorphoGraphX.org**, there are references to publications, to a “protocol” paper, and notes from courses on MorphoGraphX, but no consolidated resource. Having a single resource/manual (rather than several scattered ones) is very important. Having a single resource directs users to one set of instructions, and offers a single starting place for beginners. Additionally, if manual versions are tied to software updates, then documentation is traceable, and over time dedicated users will be able to pinpoint changes quickly that have accumulated since the last version. It may be that the internal documentation for the software is extensive, but a separate document is important still*.

We have integrated all the documentation into a comprehensive user manual. In addition to being available on www.MorphoGraphX.org, the user manual is also distributed with the software and is accessible from the “Help” menu. This ensures that users will always have access to the correct version of the documentation for their version of the software.

*The online resource with the actual software to download and manual will be critical. The current hosting at Max Planck is fine (likely to be stable), but have the authors considered someplace like GitHub or other widely used version control websites to keep their software and updates? This may make access more robust in the long term*.

We have registered the domain namewww.MorphoGraphX.org for this purpose. Internally we are using the “subversion” (SVN) source code control system, and versions of the software are identified by their SVN release number. We will put the matching source archives for all releases on www.MorphoGraphX.org.

*Related to documentation: do the authors anticipate that the community will write their own plug-ins, modules, libraries, and code for MorphoGraphX that will be shared? Will it become like ImageJ where the community contributes as much as the software platform? If so, maybe this should be built into*
*MorphoGraphX.org*
*as well*.

We encourage users to write their own plug­ins, and we hope to follow a model like Fiji, where the most useful and well tested contributions from the community are bundled in later releases of MorphoGraphX itself. We have added several example plug­ins for users to easily get started, and the Doxygen documentation for the plugin interface has been added to the “Help” menu inside MorphoGraphX. This is now mentioned in the box about www.MorphoGraphX.org.

*One way to consider providing documentation follows an example from Hadley Wickham's ggplot2. In addition to the “manual” which is cut and dry about the functions to use, there is a book and also an excellent self-published online resource. Many other R folks also publish on top of this all in the Journal of Statistical Software, or a “vignette”, to give examples of how to use packages (the* eLife *MorphoGraphX article is probably closest to a vignette)*.

*R example of documentation, ggplot2*:

*The actual manual:*
*http://cran.r-project.org/web/packages/ggplot2/ggplot2.pdf*

*The online resource*: *http://docs.ggplot2.org/current/*

*A book*: *https://books.google.com/books?id=bes-AAAAQBAJ&printsec=frontcover&dq=ggplot2:+Elegant+Graphics+for+Data+Analysis&hl=en&sa=X&ei=9U6oVJnJKsy8yQTJtoGwAw&ved=0CDIQ6AEwAA#v=onepage&q=ggplot2%3A%20Elegant%20Graphics%20for%20Data%20Analysis&f=false*

We have added a reference manual for all the functions (processes) in MorphoGraphX, and their parameters. When writing processes (plug­ins), programmers are required to provide long descriptions for the process themselves as well as all parameters. These are available within MorphoGraphX as tooltips when the mouse hovers over a process or parameter. In addition, a manual is generated on­the­fly from this information, available from the “Help” menu. This ensures that it is never out of date, and provides a way for documentation to automatically accompany plug­ins that come from other sources. After a plug­in is installed in the plug­ins folder, its documentation will immediately appear in the reference manual. We have included this process documentation at the end of the user manual for the current version of MorphoGraphX.

*3) There should be extensions/additions to the capabilities beyond what was published previously by the authors and others. Several of these applications have been hinted at in the text (for example semi-automated segmentation, extraction of 3D data from complex cell shapes (like lobed cells), tracking of subcellular elements). The detailed explanation of how to do one or two of these other analyses should be provided in the revised version*.

*Our suggestions for those one or two detailed applications are*:

*3A) Quantification of differential fluorescence levels within a cell (starting third paragraph of the subsection headed “Signal quantification” about PIN1-XFP fluorescence). Such a tool would likely be of general use. Some things to consider: Is it possible to obtain a percentage of membrane area that is occupied by PIN1-XFP compared to the whole cell? How does the software distinguish which of two neighboring cells expresses the signal? Can the software determine the overall orientation of the PIN1-XFP signal*, *as compared to the general tissue orientation (e.g. can it determine whether PIN1-XFP in root elongation zones always faces tipwards)?*

We added a supplementary figure to Figure 3 showing the quantification membrane signal and internalization of PIN1 protein, with an explanation of the quantification and its limitations.

*3B) The opportunity for subcellular analysis was hinted at, but more details are needed. For example, quantification of microtubule orientation is an important feature (*Figure 3*), but it is not explained clearly here. How is this algorithm better than the published or commercially available ones (e.g.*
[22]*)? Is it possible to get the standard deviation of the orientation (e.g. the anisotropy of MT orientation) mapped onto a tissue?*

The algorithm used in MorphoGraphX was translated directly from [6] (Nature Protocols) by one of the co­authors. The original algorithm has been improved in that the surface of cells no longer needs to be flat, it works on segmented cells, and the border is computed automatically (no additional clicking). This greatly enhances the throughput and range of datasets that the tool can be applied to.

We have added a supplementary figure to Figure 3 explaining how the microtubule orientation are computed and a discussion of the advantages of including such a process in MorphoGraphX. In addition we show how to map the anisotropy of orientation onto a tissue as the reviewer suggests.

*How might this tool be used to determine microtubule growth/shrinkage rates? Is it adaptable to track particles like vesicles or Golgi stacks? Has the software been used with data that features things moving at high velocities (e.g*. *captured with spinning disc confocal microscopy)? Where are the limitations of the software in this regard?*

It would be straightforward to add processes to MorphoGraphX to look at subcellular processes like MT extension or vesicles, or to handle other particle tracking problems in 2.5 or 3D. MorphoGraphX would be especially useful for this in cases where the tracking must be done on curved surface layers of cells.